# The Potential of Rhizobacteria to Mitigate Abiotic Stress in *Lessertia frutescens*

**DOI:** 10.3390/plants12010196

**Published:** 2023-01-03

**Authors:** Mokgadi M. Hlongwane, Mustapha Mohammed, Ntebogeng S. Mokgalaka, Felix D. Dakora

**Affiliations:** 1Department of Chemistry, Tshwane University of Technology, Private Bag X680, Pretoria 0001, South Africa; 2Department of Crop Science, University for Development Studies, Tamale P.O. Box TL1882, Ghana; 3Mamelodi Campus, University of Pretoria, Private Bag X20, Hatfield, Pretoria 0028, South Africa

**Keywords:** *Lessertia frutescens*, phytochemicals, plant-growth-promoting rhizobacteria, stress

## Abstract

*Lessertia frutescens* is a multipurpose medicinal plant indigenous to South Africa. The curative ability of the medicinal plant is attributed to its rich phytochemical composition, including amino acids, triterpenoids, and flavonoids. A literature review of some of the phytochemical compounds, particularly amino acids, in *L. frutescens* shows a steady decrease in concentration over the years. The reduction of the phytochemical compounds and diminishing biological activities may be attributed to drought and salt stress, which South Africa has been grappling with over the years. Canavanine, a phytochemical which is associated with the anticancer activity of *L. frutescens*, reduced slightly when the plant was subjected to salt stress. Like other legumes, *L. frutescens* forms a symbiotic relationship with plant-growth-promoting rhizobacteria, which facilitate plant growth and development. Studies employing commercial plant-growth-promoting rhizobacteria to enhance growth and biological activities in *L. frutescens* have been successfully carried out. Furthermore, alleviation of drought and salt stress in medicinal plants through inoculation with plant growth-promoting-rhizobacteria is well documented and effective. Therefore, this review seeks to highlight the potential of plant-growth-promoting rhizobacteria to alleviate the effect of salt and drought in *Lessertia frutescens*.

## 1. Introduction

*Lessertia (L.) frutescens* (syn. *Sutherlandia frutescens*) is a medicinal plant indigenous to South Africa (SA). The plant is potent in the management or treatment of several ailments including cancer, diabetes, epilepsy, fever, human immunodeficiency virus (HIV), stomach problems and wounds [1,2,3,4,5]. This multipurpose medicinal plant goes by numerous names, with most of them informed by what the plant is used to treat or how it looks. The most popular name is Cancer Bush or Kankerbos, inspired by the popularity of the plant as an internal cancer treatment. Other native names are Unwele in isiZulu and Lerumo la Madi in seTswana [2]. In SA, large natural populations of the plant are found in the Northern Cape (NC) and Western Cape (WC) provinces (Table 1), with the distribution extending to the Eastern Cape (EC), Free State (FS), Limpopo (L) and North West (NW) provinces.

The medium-sized plant is of Leguminosae/Fabaceae family and is characterised by small green hairy leaves and bright orange-red flowers (Figure 1). Another distinctive feature of the plant is light green, puffed up seed pods with a shade of red [1,2,3,4,5,6,7,8]. Although *L. frutescens* makes a beautiful ornamental plant, its much-needed medicinal benefits and application has rendered its use as an ornamental plant, redundant [9].

**Table 1 plants-12-00196-t001:** Distribution of *L. frutescens* natural populations in South Africa.

Location Name	Province	Reference
Burgersdorp	EC	[10,11]
Elliot	EC	[12]
Hofmeyer	EC	[12]
Jamestown	EC	[12]
King Williams Town	EC	[12]
Molteno	EC	[12]
Steynsburg	EC	[12]
Willowmore	EC	[12]
Jordaan	FS	[12]
Zastron	FS	[12]
Tubatse	L	[13]
Colesburg	NC	[10,11]
Matjiesfontein	NC	[12]
Sutherland	NC	[12]
Victoria West	NC	[10,11]
Albertina	WC	[6]
Barrydale	WC	[12]
Blouberg	WC	[6]
Calitzdorp	WC	[12]
Ceres	WC	[12]
Franskraal	WC	[6]
Gansbaai	WC	[6,10,11,12]
Heimersrivier	WC	[12]
Klawer	WC	[12]
Kruishof Farm	WC	[12]
Laingsburg	WC	[12]
Lydenburg	WC	[12]
Melkbosstrand	WC	[12]
Mount Hope	WC	[12]
Murraysburg	WC	[6]
Oudtshoorn	WC	[12]
Pearly Beach	WC	[10,11]
Prins Albert	WC	[12]
Riversdale	WC	[12]
Somerset West	WC	[12]
Uniondale	WC	[12]
Vanrhynsdorp	WC	[12]
Walkerbay	WC	[6]
Yzerfontein	WC	[6]

The monthly demand of *L. frutescens* raw material is estimated at 20 tons and cannot be met through natural populations [14]. As a result, the distribution of *L. frutescens* across SA continues to grow, mainly through cultivation. The frequent harvesting of natural populations and the worsening extreme environmental conditions has made it necessary to undertake studies towards conservation of *L. frutescens*. Cultivation of the plant in other provinces, particularly drought-stricken provinces, may result in phenotypes with compromised phytochemical profiles.

The phytochemical compounds identified in *L. frutescens* include arginine, asparagine, canavanine, flavonoids (including flavonol glycosides, Sutherlandins A–D), gamma amino butyric acid (GABA), pinitol and triterpenoids (including triterpenoids glycosides, Sutherlandiosides A–D) [1,2,5,6,7,8,15,16]. The mechanism by which these phytochemicals facilitate human health is not yet fully understood. However, Colling et al. [7] and Van Wyk et al. [17] suggested that these compounds maybe work synergistically to promote human health, a speculation which is contradicted by findings of [10] and [11]. Their results attributed the superior anticancer activities evident in the extracts of *L. frutescens* samples collected in the NC to Sutherlandiosides. Meanwhile, the exceptional antioxidant activities of the WC samples were attributed to the Sutherlandins, which are only present in the WC samples. Of the many phytochemicals in the plant, canavanine has received the greatest attention from researchers, owing to its ability to inhibit tumor proliferation [7,18]. Katerere and Eloff [4] also attributed the antiviral activity of *L. frutescens* to canavanine.

It is without doubt that the biological activities (anticancer, antibacterial, antiviral and antioxidant) of *L. frutescens* depend on the identity and concentrations of phytochemicals present [19]. However, total phytochemical content is not always directly correlated to biological activities [11]. This suggests that in some instances. individual phytochemicals are responsible for certain biological activities, just as [10] and [11] demonstrated in their studies. The observed variability in the biological activities of *L. frutescens* among authors may be attributed to disparities in the phytochemical composition, which may have been brought about by the prevailing abiotic stress. For example, drought was found to reduce the concentration of canavanine in *L. frutescens* [7], suggesting deleterious effect of drought on the concentration of phytochemicals. This was contrary to the general notion that abiotic stress improves the efficacy of medicinal plants by facilitating the accumulation of phytochemicals in medicinal plants [20]. The complex interaction of plant species with environmental variables makes it difficult to generalize the influence of abiotic factors on the biological activities of medicinal plants, emphasizing the need for species-specific studies in response to such factors [19].

Aside from the effects of abiotic stress factors such as drought, overall plant health is influenced by soil microbial communities within the rhizosphere [21]. Beneficial microbes in the rhizosphere promote plant growth through several mechanisms that include the solubilization or release of nutrient elements from sparingly available sources [22]. For example, in legumes such as *L. frutescens*, Nitrogen (N_2_) fixation in association with plant-growth-promoting rhizobacteria called rhizobia can supply the plants N_2_ requirements for growth [13]. Broadly put, plant-growth-promoting rhizobacteria (PGPR) are soil bacteria with the ability to regulate plant growth and development directly or indirectly [22,23]. Taken together, the activities of these beneficial microbes may enhance the relative concentrations of phytochemicals in plants, even under extreme conditions [23]. However, exploitation of PGPR in the mitigation of negative effects associated with abiotic stress in medicinal plants is understudied [23]. This is despite the overwhelming literature that validates their mitigation capabilities [24,25]. In addition to their efficacy to mitigate the deleterious effects of abiotic stress, PGPRs are an environmentally friendly alternative to the chemically-based commercial fertilizers. Chiappero and colleagues [24] employed PGPR to alleviate the effect of drought on *Mentha piperita*. Their intervention resulted in enhanced total phenolic content and antioxidant activities. In addition, a significant decrease in proline, the stress-induced amino acid, was noted in the plants treated with PGPR. This indicates that the plants inoculated with PGPR were not strained by the drought. Meanwhile, in a similar study, inoculation of drought-stricken *Mentha piperita* with PGPR resulted in increased essential oil yield, plant biomass, and menthol content [25]. Various and intricate mechanisms are employed by the PGPR in the alleviation of abiotic stress. For instance, *Bacillus subtilis* engaged in the production of auxin indoleic acetic acid and the suppression of ethylene levels when it was introduced to *Bassia indica* stressed with salt [26].

This paper explores the potential of using plant growth promoting rhizobacteria to alleviate the effects of drought and salt stress on the phytochemical profile of *L. frutescens*, an important medicinal plant in the Leguminosae family.

## 2. Methodology

A systematic literature search was conducted to find studies focusing on the effect of abiotic stress on the phytochemical content or biological activities of *Lessertia frutescens* extracts or the use of plant-growth-promoting rhizobacteria in the plant. The Preferred Reporting Items in Systematic Reviews and Meta-Analyses (PRISMA) checklist was followed to ensure reliability and minimize publication bias [27]. Four reputable web-based search engines, Google Scholar, PubMed, Scopus, and Web of Science, were used to explore literature repositories. The Google Scholar search served as a gray literature tool since it is an effective search engine to locate dissertations, theses and projects by other organizations [28]. In all searches, the main keywords ‘*Lessertia* OR *Sutherlandia frutescens*’ were used in conjunction with secondary keywords (drought OR salinity OR salt stress OR abiotic stress) AND (metabolites OR phytochemicals OR rhizobia OR rhizobacteria OR plant growth promoting rhizobacteria OR inoculum OR inoculant). The search was limited to title, abstract and keywords. However, the reference list of the articles obtained through a database search was further explored to find articles that may match the search criteria. This was done to ensure that all published and unpublished articles are included in the review. The time range for the search was set between January 2010 and May 2022.

After the search, duplicates and irrelevant articles were removed. Abstracts of the articles were read to establish their focus. Therefore, articles focusing on other *Lessertia/Sutherlandia* species besides *frutescens* and topics beyond the secondary keywords were excluded. Papers on drug transportation, seed germination, epistemology and case studies were also excluded. However, studies that focused on *L. frutescens* and phytochemical content, *L. frutescens* and methods to enhance phytochemical content, and *L. frutescens* and drought or salt stress were all included. In addition, articles that related the microbial activities of *L. frutescens* to phytochemicals were included. Articles that studied the anti-microbial activities, anti-cancer and anti-diabetic activities of the plant extracts without linking the activity to specific phytochemicals were excluded.

The literature search yielded the following number of articles from respective search engines: Google Scholar (*n* = 1070), PubMed (*n* = 19), Scopus (*n* = 34), and Web of Science (*n* = 28). After meticulous selection of the articles using the inclusion and exclusion criteria, and removal of duplicates, only 29 articles were considered for the review. A summary of the results is outlined in Figure 2, below. The 29 included 19 research articles, 5 review articles, 4 theses and 1 clinical trial.

## 3. Phytochemical Composition of *Lessertia frutescens*

Increased interest regarding the health benefits of *L. frutescens* has encouraged researchers to look into the phytochemical composition of the plant. To date, a variety of compounds has been isolated from mainly the aerial part of the plant and are reported to provide the basis for the plant’s multipurpose medicinal benefits [1]. So far, the phytochemicals isolated include, but not limited to amino acids, flavonoids (including flavonol glycosides), pinitol and triterpenoids (including triterpenoids glycosides) [1,2,5,6,7,8,15,16,29].

### 3.1. Amino Acids

Apart from being the building blocks of proteins, amino acids play a key role in the growth, development and well-being of humans [30]. The molecules consist of two broad categories, essential and non-essential. The term non-essential amino acids does not mean the amino acids are not crucial, but it refers to amino acids that are mainly synthesized by body cells, which then need to be provided for through diet. Meanwhile, the essential amino acids are not synthesized by body cells and should be compensated for through diet [30]. The study conducted by Mncwangi and Viljoen [12] on wild and cultivated populations of *L. frutescens* from various locations revealed that amino acid content ranged from 10 to 15%, and 60% of the total amino acids, comprising asparagine, alanine and proline. In this study, about 100 mg of dried homogenized leaves was extracted with 50% acetonitrile and 0.1% formic acid mixture prior to analysis. The findings identified asparagine, alanine, and proline as major amino acid constituents of *L. frutescens*. Although not part of the major amino acids of *L. frutescens*, canavanine and gamma aminobutyric acid (GABA) enjoy scientists’ attention, due to their notable health benefits [17,31,32].

A comparison of the concentration of amino acids in natural populations of *L. frutescens* is depicted in Table 2. The two studies by Mncwangi and Viljoen [12] and Moshe [33] were conducted 14 years apart. The results highlight a significant depreciation in overall amino acid concentrations over the years. The authors are cognizant that the evaluation should be considered with caution, since the samples were not collected from the same location. Although both authors used leaves for analysis, Mncwangi and Viljoen [12] used an acetonitrile: formic acid mixture, while Moshe [33] used ethanol to prepare extracts. In another study, Shaik and colleagues [15] reported GABA and canavanine content to be 3.48 and 0.08 mg/g, respectively, in the methanolic extracts of leaf samples, confirming the depreciation of amino acid content as noted above. These results were consistent with the 3 mg/g GABA and 0.4 mg/g canavanine content, wherein methanolic extracts of pulverized leaves were used for an analysis [5].

#### 3.1.1. Alanine

Alanine (Figure 3a), commonly known as L-alanine, is a non-essential amino acid [34] with benefits for humans. According to Araujo and colleagues [35], alanine is instrumental in regulating pancreatic β-cell physiology to minimize lifestyle-induced obesity. For example, it is used in the regulation of healthy blood sugar levels through gluconeogenesis. Gluconeogenesis is the process of producing glucose molecules in the liver [36]. Thus, avid exercisers benefit greatly from the involvement of alanine in gluconeogenesis. Mncwangi and Viljoen [12] reported alanine concentrations in *L. frutescens* ranging from 0.37 to 1.67 mg/g, while Moshe [33] observed concentrations of up to 10.7 mg/g. A significant depreciation in the phytochemical is evident in the results.

#### 3.1.2. Asparagine

Asparagine (Figure 3b) is an amide of amino succinic acid which is partially soluble in water. Asparagine comprises a standard nitrogen: carbon ratio of 2:4. This special trait makes the molecule a suitable candidate for the storage and transportation of N_2_ in living organisms [37]. In a very recent study, it was suggested that asparagine reduces viral replication in human cells. This discovery makes the phytochemical a potential agent for antiviral development [38]. Soluble asparagine accumulates in almost all parts of plants. It has also been noted that the accumulation of this compound may occur as a result of abiotic stress [37]. However, Dell’Aversana and co-workers [39] reported contrasting observations. They observed a significant decrease in the asparagine content of *Hordeum vulgare*, which was subjected to salt stress. So far, the influence of abiotic stress on the accumulation of asparagine has not been explored in the case of *L. frutescens*. The available literature suggests that the asparagine concentration in *L. frutescens* ranges between 3.00 and 61.0 mg/g [12,33]. The contradicting reports on the content of asparagine in plants subjected to abiotic stress highlights the importance of seeking to study specific plants before making conclusions, in order to avoid inaccurate claims.

#### 3.1.3. Canavanine

Canavanine (Figure 3c) is a non-protein amino acid commonly present in leguminous plants. The metabolite exhibits antagonistic characteristics against arginine and may inadvertently substitute arginine molecules during protein synthesis [40]. This phenomenon leads to abnormal three-dimensional protein molecules, also called canavanyl proteins [20,40]. The substitution of arginine by canavanine is not a favorable biosynthesis pathway as it may interrupt enzyme activities and subsequently obliterate the conformation of proteins [40]. Relatively high concentrations of canavanine are detected in the seeds of leguminous plants, while moderate levels are found in their aerial parts [20,40]. For example, canavanine constitutes a major proportion of the amino acids in alfalfa seeds.

Canavanine is known for its superior antitumor properties [18]. Research into this claim began decades ago, with the in vivo experiments on mammals taking place in the early 1980s. The study by Green et al. [41] assessed the anticancer properties of canavanine in leukemic mice infected with L1210. Their findings indicated that high doses of canavanine injections to the mice deactivate cancerous cell proliferation. In addition, the treatment increased the life span of the mice by 44%. The anticancer activity and mechanism of the phytochemical was reviewed by Bence and Crooks [42]. In addition to the antitumor properties, canavanine is reported to provide protection for plants against pathogens and predators [18,40].

Canavanine is among the most-studied amino acids in *L. frutescens*. Of interest is that Mncwangi and Viljoen [12] observed a huge variation in the concentration of some amino acids, including canavanine, in propagated *L. frutescens* plants. To date, the concentration of canavanine in the aerial part of *L. frutescens* ranges from 1.00 to 18.0 mg/g [5,12,15,33]. Such discrepancies in the phytochemical composition of medicinal plants are unfavorable, lest they result in challenges for formulation of standardization protocols. Therefore, practical and sustainable interventions on the cultivation level to optimize phytochemical compounds in the plant are necessary.

#### 3.1.4. Gamma Aminobutyric Acid

Gamma aminobutyric acid (Figure 3d) is a four carbon non-protein amino acid essential biochemical, commonly found in microorganisms and plants [43,44,45] Various types of foodstuffs contain low levels of GABA, with fermentation causing an escalation in its concentration [46]. It is suspected that lactic acid is a precursor for the phytochemical [47]. Gamma aminobutyric acid was initially regarded as just a phytochemical. However, decades after its discovery, speculations that the compound could be a signaling molecule surfaced. Extensive research to investigate the suggestion commenced and still continues [45,48]. Unlike proline, GABA is a flexible compound and can assume various structural conformations, including a cyclic shape. Gamma aminobutyric acid induces anti-abiotic stress defenses in plants by preventing reactive oxygen species (ROS) formation, redox imbalance and cell death [20].

Of all the amino acids reviewed in this study, GABA is the second-most beneficial to humans, after canavanine. It is highly esteemed for its inhibitory neurotransmitter properties in the central nervous system [47]. Furthermore, Inoue et al. [49] reported that blood pressure reduction was witnessed in humans who consumed GABA. In addition to these benefits, GABA can induce relaxation and reduce anxiety. These benefits may be the reasons why GABA is used as a fortifier for various foodstuffs in Japan [47].

*Lessertia frutescens* natural populations contain GABA content, which is relatively close to the minimum range. For instance, Moshe [33] and Tai and partners [5] reported GABA concentrations of 0.045 and 0.029 μmol·g^−1^, respectively. According to [45] the concentration of GABA in plants shoots up under extreme environmental conditions. Therefore, under extreme environmental conditions, the increase in GABA may be accompanied by a reduction in the concentration of L-glutamate, because it is a precursor molecule for GABA. Again, this claim needs to be ascertained in the case of *L. frutescens*. As a result, GABA is often used to circumvent adverse effects of extreme environmental conditions. Priya and co-authors [46] demonstrated how GABA application can benefit the crop industry through alleviation of heat in mung beans. The authors observed a decline in endogenous GABA contents of the plants subjected to heat stress. However, the application of exogenous GABA to the plants resulted in improved plant parameters when compared with just the heat-stressed mung beans. Meanwhile, Yang et al. [50] soaked peach fruits in GABA solution to develop resistance to chilling injury due to the activation of antioxidant enzymes and the maintenance of higher energy level status in the fruits. In the most recent study, GABA was used as a drought stress suppressant in two variants of black pepper plants [51]. Physiological parameters of pretreated pepper plants were better than untreated plants, when both were subjected to drought stress.

#### 3.1.5. Proline

Proline (Figure 3e) is a unique proteogenic amino acid with its γ-amino acid group positioned as a secondary amine. The compound is characterized by a rigid, cyclic structure, which restricts conformational flexibility [52,53]. The fact that proline is considered a non-essential amino acid in humans unless they are injured has almost disguised its indispensable value in neonates [54]. Interesting discoveries were made in studies that involved pig neonates, which could be extrapolated to human neonates, since similarities in the two were validated [54]. In one study, a remarkable demand for proline, used in protein synthesis, was noted during pre- and postnatal periods [55]. Meanwhile, Brunton et al. [56] indicated a huge dependability of protein synthesis in neonates, on parenteral proline supply.

Proline plays a vital role in plant primary metabolism [57]. Its compatible solute characteristics are remarkable in dehydrated seeds and pollen, where it combats cellular structure degradation [52]. It has also been reported that the metabolite plays a critical role in plant pathogen interactions, programmed death and development [53]. Proline accumulation in plants is attributed to their exposure to stressful environmental conditions such as salt, drought, ultraviolet (UV) radiation, extreme temperatures, and toxic metals [57]. This accumulation is apparently at a pace that is relatively faster than any other amino acids. Hence, other researchers proposed its exploitation as a tool to develop irrigation protocols for plants and select drought resistant species. However, in the case of *L. frutescens* there is no data available to support or refute whether the metabolite accumulates under extreme environmental conditions. The concentration of proline in *L. frutescens* natural population aerial parts ranged between 0.32 and 36.8 mg/g [12,33]. High concentrations of proline are purportedly found in the reproductive organs of plants when compared to other plant parts. The variability extends to plant leaves, where young ones contain relatively high concentrations when compared to the mature counterparts [52].

### 3.2. Pinitol

Pinitol (Figure 4) is a cyclitol which was first isolated from *Pinus monticola*, from which the name was derived. It occurs naturally in plants, especially of the Leguminosae family [58]. The concentration of pinitol in plants may vary in response to abiotic stress. Considering that pinitol is among the phytochemicals that are responsible for *L. frutescens’* medicinal properties, it does not come as a surprise that several studies [12,15,33] have documented its valuable health benefits in humans. These include antidiabetic [59,60], antioxidant [61] and anticancer [62,63] capabilities. A significant improvement in the condition of a liver was noted in the patients of non-alcoholic fatty liver disease, when pinitol was administered. Pinitol induced the reduction of oxidative stress and accumulation of fatty acids, in order to effect the improvement [61]. Meanwhile, it induced cytotoxicity in human leukemia cells, MOLT-4. In this case, the dose-dependent cell apoptosis was achieved through the generation of reactive oxygen species [62]. Thus, a reduction in the level of this phytochemical in plants may lead to compromised efficacy.

*Lessertia* consists of several subspecies that are very similar, and that may be difficult for laymen to distinguish [17]. Moshe [33] determined the concentration of pinitol in *Lessertia frutescens* in subspecies *microphylla* only, and found it to be 14.4 mg/g. Meanwhile, Shaik, Singh and Nicholas [15] found pinitol to be the most abundant in the field-grown leaves of *L. frutescens* at a concentration of 18.17 mg/g, while Mncwangi and Viljoen (2012) reported concentrations ranging between 0.02 and 1.32 mg/g. The field leaves that Shaik and colleagues [15] used in their study were sourced commercially, rendering it difficult to find more details about their origin. Knowledge of the exact source or location of the samples would provide insight into how the environment may affect pinitol content and allow informed deliberations on the variation of pinitol that is evident. Previously, it was reported that variable geographical locations may result in different metabolite profiles [10]. These results also demonstrate that the concentration of pinitol fluctuates vastly due to factors such as storage, cultivating and harvesting season and abiotic stress [15].

### 3.3. Triterpenoids

Triterpenes are naturally occurring C30-terpene compounds characterized by isoprene subunits and diverse structures. Terpene compounds are classified into monoterpenes (C10), sesquiterpenes (C15), diterpenes (C20), sesterpenes (C25), triterpenes (C30), and tetraterpenes (C40), depending on the number of carbons in their structure [64,65,66]. Although often interchanged with triterpenes, triterpenoids are slightly different from triterpenes. Triterpenoids are a collective name for all triterpenes, including natural degradation products, natural and synthetic derivatives, and hydrogenation products [64,65,67]. There are more than 20,000 identified triterpenoids and the number continues to grow due to the heightened interests in the compounds because of their bioactive capability [64,66]. Moreover, there are two main groups of triterpenoids, tetracyclic and pentacyclic triterpenoids, with the latter group being the largest, while monocyclic, bicyclic, tricyclic, hexacyclic and acyclic triterpenoids also exist [66,67]. The structural diversity of triterpenoid compounds is attributed to the enzyme oxidosqualene cyclase process which facilitates the rearrangement of triterpenoids scaffolds to form distinct structures [66].

#### Triterpenoid Glycosides

Triterpenoid glycoside is a general term for a natural product, a triterpenoid in this case, attached to glycone compounds [68]. Researchers continue to isolate and identify novel triterpenoid glycosides from plant extracts [69,70]. Triterpenoid glycosides act as self-defense agents for plants. For example, Kubanek et al. [71] established that triterpenoid glycosides served as deterrents for predators of marine sponges. The compounds also exhibit anti-proliferative properties on cancer cells [72]. Fu and co-workers [73] isolated and identified four novel triterpenoid/cycloartane glycosides, Sutherlandiosides A–D (Figure 5), from *L. frutescens* extracts. The group also established that Sutherlandioside B was the major compound of the four. Furthermore, Fu and co-workers [73] proposed for Sutherlandioside B to be used as a biomarker for *L. frutescens*, a proposal which was refuted by Albrecht and partners [6] on the basis that the compound is not detectable in all populations and it has not been certified as the main active ingredient. In addition, Brownstein and co-workers [29] proposed that Sutherlandioside B may have caused a reduction of corticosterone in rats that had chronic immobilization stress, validating it as a potential anti-stress agent in humans.

To date, there has not been a conclusive investigation on the biological activities of the individual phytochemicals of Sutherlandiosides A–D, partly due to complications associated with attaining their pure compounds. Should pure isolates of these compounds be accessible, accurate speculations into their individual biological activities would be achieved. A successful attempt was however made to isolate and purify Sutherlandioside B [29]. When an antiviral activity assay was conducted for the isolated Sutherlandioside B against Epstein–Barr virus (EBV), very little activity was evident. However, the detected antiviral activity was lower than that of the positive control employed in the study. Moreover, unpublished data from Fu [74] indicated no antimicrobial activity on the four triterpenoid glycosides, despite increasing their concentration up to 20 µg/mL. The results for the little to no antiviral activity of Sutherlandiosides found in *L. frutescens* should however not deter the interests in their health benefits. Rather, it should spur researchers to look into other biological activities of the compounds, because their presence in *L. frutescens* is associated with superior anticancer activities [11].

Albrecht and partners [6] discovered two types of *L. frutescens* ecotypes in samples collected from the Cape, those that contain detectable amounts of Sutherlandiosides B (NC), and others without the compound (WC). The samples for this study were prepared by extracting the whole plant with an acetonitrile: formic acid mixture. The authors speculated that the two types of plants (with and without Sutherlandioside B) were *incana* and *microphylla* subspecies. However, they do discount the possibility of variation due to different origins and environmental conditions. According to Zonyane and partners [10] Sutherlandiosides B and C were detected in the ethanolic extracts prepared from the leaves of *L. frutescens* collected in the NC, while WC samples did not contain any of these phytochemicals. These findings corroborate what Albrecht and partners [6] speculated, insinuating that plants from the two provinces are of different subspecies. Studies that explored the variation in the Sutherlandiosides A–D profile to see if they can be linked to specific subspecies concluded that subspecies *L. microphylla* is characterized by the presence of Sutherlandiosides B and D [75].

### 3.4. Flavonoids

Flavonoids are diverse polyphenolic class of compounds and secondary metabolites, occurring naturally in plants. These compounds consist of 15-carbon atoms (Figure 6) fashioned in three rings as C_6_-C_3_-C_6_ [16,76]. To date, the highest number of flavonoids isolated is 9000 [76], with other studies citing 6500 [77] and 4000 [78]. Thus, the numbers continue to grow due to increasing interest in the health benefits conferred by these compounds when present in human and animal diets. The biological activities possessed by the phenolic compounds include antioxidant activity, weight management, protection against cardiovascular diseases, anti-allergy properties, antibacterial activity, anti-inflammatory effects, anticancer properties and age-related neurodegenerative disease prevention [76,79]. Flavonoids cannot be synthesized by animals. However, plants, particularly higher plants, synthesize them from phenylamine through the shikimic acid pathway [79].

Flavonoids are reported to give plants their distinct color, flowers special aroma, and fruit characteristics that attract pollinators [77]. They also play a vital role in seed germination, growth and development of seedlings [77]. Pourcel et al. [80] highlighted the ability of flavonoids to alleviate biotic and abiotic stress in plants. It is reported that the compounds act as UV-radiation filters, thereby minimizing the impact of abiotic stress [80].

Flavonoids are categorized based on the substrate that attaches to the C_3_ ring of the fundamental C_6_-C_3_-C_6_ skeletal structure. Some of the classes of flavonoids include chalcone, flavones, flavonols, flavanones, anthocyanins, and isoflavonoids [76,79]. Most of these groups have established significant commercial value in the market [79]. They exist commonly in a glycosylated form [76,77], and four of them were discovered by Fu and co-authors [16] in *L. frutescens*.

#### Flavonol Glycosides

Flavonol glycosides are flavonoids with a glycoside substrate attached to the C_3_ ring of the flavonol compounds. To date, of the six novel flavonol glycosides in *L. frutescens* discovered and discussed by Van Wyk and Albrecht [17], only four were successfully isolated and identified. Fu and colleagues [16] established the structures of the four compounds and named them Sutherlandins A, B, C and D (Figure 7). These compounds are quercetin and kaempferol derivatives [6]. Although they are not major compounds of the medicinal plant, they may be used as its biomarker or to distinguish origins of *L. frutescens* raw materials. The findings of Zonyane and co-workers [10] provide a basis for their use to predict the origin of raw materials, since only samples from the WC province of SA contained Sutherlandin B, while samples from the EC and NC provinces contained both Sutherlandin A and D. Meanwhile, Acharya and colleagues [75] used the presence of these glycosides to differentiate between subspecies. For example, Sutherlandin B’s presence was noted in *L. frutescens*, while Sutherlandin A was found in *L. microphylla*.

Although flavonol glycosides may exhibit some of the general health benefits of flavonoids, it is imperative to investigate compound-specific activities. This would only be possible if pure compounds of these glycosides were readily accessible. Thus, the work of Chen et al. [81] could have an application, provided the yields are improved. Chen and co-workers (2017) reported a yield of 110.4 mg per 0.90 g plant extract, which the authors regard as low and wish to enhance.

## 4. Variations in Phytochemical Content of *Lessertia frutescens* Due to Abiotic Stress

Despite there being evidence that *L. frutescens* development varies depending on abiotic stress factors, there is inadequate data on the plant’s response to various stresses. A blanket approach to plants’ response to abiotic stress is strongly discouraged, since plant species behave differently [19]. Therefore, an investigation on the impact of drought and salt stress on *L. frutescens* and potential tailor-made interventions to circumvent the impact are crucial. In addition to abiotic stress, biotic stress, seasonal variations, plant age, geographical location, irrigation regimes, storage methods and drying protocols were cited as other factors responsible for variation in the biological activities or phytochemical composition of medicinal plants [19]. These sometimes lead to reduced phytochemical content, low biomass and compromised biological activities, which in turn pose a risk to consumers of medicinal plants. Overdose or underdose may occur since consumers measure doses based on the amount of raw material, while being oblivious of the variations in the biosynthesis of compounds. Although underdosing may not be as detrimental, it would render the medicine ineffective, while overdosing may have much more harmful effect [19].

Colling and colleagues [7] studied the effect of nitrogen supply, drought and salt stress on the canavanine concentration in *L. frutescens*. A 1.9 g L^−1^ KNO_3_ and 1.65 g L^−1^ NH_4_NO_3_ dose resulted in a four-fold increase in the concentration of canavanine when compared to half the dose. Although the treatment resulted in an increase in canavanine, it should be noted that application of chemical-based fertilizers such as KNO_3_ is unfavorable and discouraged since it is damaging to the environment. In the same study, treatment of plants with 100 mM NaCl, led to a slight increase in canavanine. A slight reduction in canavanine was however apparent in the plants treated with 3% polyethylene glycol (PEG), to simulate drought stress. Thus, it is highly probable that phytochemicals in the plant respond differently to different stressors, and this therefore highlights the need for their continued investigation. This is another reason why investigation into the hypothesis that abiotic stress has an impact on the phytochemical profile of *L. frutescens* needs to be undertaken.

Zonyane and colleagues [11] examined samples of *L. frutescens* from different geographic locations by screening them for novel phytochemicals followed by the determination of antioxidant and anticancer activities. Their investigation was two-fold. They screened the samples to establish which novel phytochemicals were present, then they determined antioxidant and anticancer activities. Interestingly, the WC populations were the only ones that contained Sutherlandin B. This was not an isolated case of the flavonol glycosides being a distinguishing factor for WC chemotypes, since Zonyane and co-workers [10] had previously detected Sutherlandin B in samples collected from the WC. Sutherlandin A, C and D, and Sutherlandioside A, B, and C peaks were observed in the NC chemotype. Again, Sutherlandioside B and C were characteristic of the NC samples [10]. In addition, the chemotype that exhibited the most effective antioxidant was from the WC province, whilst superior anticancer activity was evident in the NC chemotype. The NC weather is considered harsh, with little precipitation in summer and extreme drought that is exacerbated by the excessively hot summer seasons [82]. Unlike the NC, the rainfall season of the WC is around winter and the precipitation rate has significantly depreciated in recent years [83]. Thus, the two provinces are distinct in their climate and will most probably exert different abiotic stresses on plant species. The unique results for the biological activities of the samples from the NC and WC provinces partly validate the theory that triterpenoid glycosides are anticancer agents, while flavonoids are the most effective antioxidants [11]. It has also been shown that significant variations occur in *L. frutescens* samples from different locations.

These findings underscore the urgent need for a sustainable, environmentally friendly, and effective intervention to minimize the effect of drought and salt stress [84]. Not only will the proposed solution benefit the commercial medicinal plants industry by increasing medicinal plants yield and value but it will also improve the lives of consumers.

## 5. Growth and Phytochemical Enhancement in *Lessertia frutescens*

To the best of our knowledge, no information could be retrieved from the literature on the alleviation of drought or salt stress in *L. frutescens*. However, several studies focused on methods to improve *L. frutescens* growth [85], biomass [86], total phenolics [85], total flavonoids [85,87], amino acids [87], and terpenoids [87]. Although the approaches did not employ drought- or salt-stressed plants, they could provide valuable information on how *L. frutescens* responds to the different interventions, information that could be employed during the formulation of a solution to the abiotic stress impact on *L. frutescens*.

Makgato and partners [88] explored the application of a commercial plant-growth-promoting rhizobacteria (PGPR) inoculant to enhance phytochemical profiles in *L. frutescens*. Their study revealed an improvement in the total phenolics and antioxidant activity and a minor change in biomass and nitrogen fixation. The insignificant change in biomass and nitrogen fixation incites uncertainties surrounding the efficacy of the inoculum employed in their study. The inoculum used was formulated from *Bradyrhizobium* and *Rhizobium* microsymbionts. The choice of the inoculum is questionable, since Gerding and colleagues [89] already established that *Lessertia* species are nodulated by *Mesorhizobium* symbionts. Moreover, Gerding and others [90] demonstrated a potential competition between *Mesorhizobium* and *Rhizobium* species when *L. frutescens* was inoculated in soil populations where *Rhizobium* exists as native symbionts. Therefore, the inoculation of *L. frutescens* in the investigation may have been successful due to the inferior selectivity of *L. frutescens*, hence, the unsatisfactory results. Further investigations should consider using indigenous symbionts of *L. frutescens*.

Masenya and colleagues [13] compared the efficacy of native rhizobia strains for *L. frutescens* and commercial inoculum formulated from *Bradyrhizobium* and *Rhizobium* symbionts. The native symbionts exhibited superior plant growth parameters and symbiotic effectiveness when compared to the commercial strains. The native bacterial symbionts used were isolated from root nodules of *L. frutescens*, which were collected in the L province. Meanwhile, the symbionts that Gerding and colleagues [89] studied originated from the NC and WC provinces. Valuable insights would be gained if the native symbionts were sequenced, or their genus established. As anticipated, the commercial inoculum has performed worse than the native symbionts, validating the suggestion that *Bradyrhizobium-* and *Rhizobium*-based symbionts are ineffective symbionts for *L. frutescens*.

The use of plant-growth-promoting bacteria to enhance growth and biological activities in medicinal plants subjected to drought and salt stress has been explored for several plants, but to date no attempt was made on *L. frutescens*. The PGPR usually promotes growth by two mechanisms, direct and indirect. The direct mechanism includes nitrogen fixation, which is signified by the formation of root nodules in plant members of the Leguminosae in association with rhizobia.

Raselabe [85] investigated the effect of pruning and commercial fertilizers (NPK) on the growth, phytochemical properties, and antidiabetic activity of *L. frutescens*. Pruning is a conventional plant-growth-promoting method that entails the removal of certain parts of a plant. The technique is employed to regulate growth, control branching, compensate for transplant damage, promote blooming and to stimulate plant wellbeing [85]. According to their findings, pruning resulted in enhanced total phytochemical compounds, which could have been due to the plant responding to the shock signal by producing phytochemicals, some which act as defense agents [85]. The phytochemicals were qualitatively determined, meaning it is not clear which specific ones increased. These findings warrant further evaluation. For instance, although the phytochemicals increased, a correlation between the concentration of the phytochemicals and antidiabetic activity was not evident. This can be because the phytochemical that increased was not implicated in antidiabetic activity. In a similar study, the effect of phosphate and ammonium-based fertilizers on *L. frutescens* was assessed. Phosphate-based fertilizer was toxic to the medicinal plant, while the application of ammonium sulphate fertilizers resulted in an improved chemical profile [91]. However, the identity of the specific phytochemicals was not established. Both these studies involved the use of commercial chemical fertilizers, which are discouraged due to their deleterious effect on the environment and soil quality (Liu et al., 2010).

The study by Grobbelaar et al. (2014) which focused on establishing the effect of strigolactones and auxins on *L. frutescens* reported promising results. Plants that were treated with 1-naphthalene acetic acid (NAA) and Nijmegen demonstrated that the treatment favored accumulation of amino acids (alanine, arginine, proline, etc.), flavonoids and terpenoids (Sutherlandioside B). Although the use of these chemicals may enhance phytochemical accumulation in *L. frutescens*, the potential environmental footprints and issues of sustainability need to be addressed.

## 6. PGPR for Alleviation of Drought and Salt Stress in Medicinal Plants

Plant-growth-promoting rhizobacteria are a group of bacteria that colonize the rhizosphere and promote plant growth through direct and indirect mechanisms [92,93]. The direct mechanism includes nitrogen fixing, production of phytohormones, potassium and phosphorus solubilization [93]. Meanwhile, the indirect mechanism may entail the prevention of plant disease and induction of plant defense against pests and diseases. Some reports suggest that plants produce more superoxide dismutase, catalase peroxidase, glutathione reductase and ascorbate peroxidase enzymes to facilitate salt and drought stress tolerance [22,94]. According to literature, there are about 10^8^ to 10^9^ bacteria per gram of healthy soil and the number may drop as low as 10^4^ in drought-stricken soils [93]. The distribution of bacteria in soils is not even, but instead the rhizosphere contains a large fraction of bacteria when compared to the bulk soil. This is because plant roots exude amino acids, sugars and organic acids that serve as food for several bacteria, making the rhizosphere an appealing residence [93,95]. The properties that distinguish potential PGPR include, but are not limited to, the ability to colonize the root surface, compete, proliferate and surpass other microorganisms in the soil, and promote growth [23].

Alleviation of drought and salt stress in medicinal plants through inoculation with PGPR is well documented and effective [22,26,94,96]. The PGPR facilitates abiotic stress tolerance through direct and indirect mechanisms that are mainly dependent on the genotype and the age of the plant [93,97,98]. The direct mechanism includes biological nitrogen fixing, solubilization of nutrients, expression of 1-aminocyclopropane-1-carboxylate (ACC) deaminase and phytohormones such as indole-3-acetic acid (IAA), absciscic acid (ABA), cytokinins, etc. Plant-growth-promoting rhizobacteria may employ one or more of these pathways, but most PGPRs possess only a few [98]. Plant-growth-promoting rhizobacteria which produce IAA and ACC deaminase are reported to induce abiotic stress tolerance through ion homeostasis. For example, a study by Wang and colleagues [99] reported an increase in K uptake by pea plants when they were inoculated with ACC deaminase PGPR.

The indirect mechanism hinges on controlling the phytopathogens through the production of volatile organic compounds (VOCs), siderophores, antibiotics, hydrogen cyanide, etc. [97]. Production of VOCs benefits host plants by facilitating the extermination of phytopathogens. Additionally, VOCs may initiate the production of plant genes responsible for reactive oxygen species-scavenging enzymes such as glutathione reductase, superoxide dismutase and catalase. As a result, the production of ROS-scavenging enzymes will minimize the negative effects of ROS on plants under abiotic stress [93].

Golpayegani et al. [94] demonstrated how *Pseudomonades* sp. and *Bacillus lentus* induced salt stress tolerance in basil plants. The treated plants exhibited improved antioxidant activities and increased phytochemicals when contrasted with untreated plants. There was also a noticeable improvement in their biomass. Meanwhile, salt-stressed *Indian bassia* was inoculated with *B. subtilis* to investigate whether the strain would ameliorate plant development. The strain successfully employed a mechanism that produces auxin, IAA and curbs the production of stress ethylene levels to minimize the effect of salt stress [26]. The treatment resulted in enhanced biomass, total lipid content and photosynthesis pigments in the plant. Inoculation of drought stricken *Hyoscyamus niger* with *Pseudomonas putida* and *P. fluorescens* revealed the stress tolerance ability of the two strains. In general, the inoculation minimized the adverse effects of drought on plant growth parameters. A significant increase in superoxide dismutase and peroxidase enzymes was noted, particularly in the roots and leaves. Enhanced proline accumulation was noted in plants that were inoculated with only *P. fluorescens*. Meanwhile those that were treated with *P. putida* were characterized by an increased alkaloid content.

These studies propose exploitation of PGPR as potential, viable and sustainable approach to alleviate drought and salt stress in medicinal plants [22,26,94,96].

## 7. Conclusions and Prospects

*Lessertia frutescens* is an invaluable medicinal plant in SA and the whole globe. Measures to improve its efficacy will be of significance to health sector and socioeconomic status of the society. High and consistent phytochemical profiles of *L. frutescens* will eliminate the incidences of overdose or underdose due to fluctuating phytochemical profiles caused by abiotic stress effects.

Although it is not obvious, the depreciation of the amino acid contents in *L. frutescens* over the years may be because of the persistent drought in SA. Any reduction in the amino acid profile would be detrimental to the medicinal capabilities of the plant because previous research has established their contribution to the efficacy of *L. frutescens*. This prospect challenges researchers to devise robust cultivation protocols which are not prone to unfavorable alterations of *L. frutescens* due to abiotic stress.

Of the several measures available to improve the phytochemical profile of *L. frutescens*, the use of PGPR is a more sustainable and cost-effective method as it exploits biological processes. Furthermore, the use of PGPR offers an environmentally friendly alternative when compared to the common chemical fertilizers, which are harmful to the environment. Exploring *L. frutescens* endemic soils in drought-stressed and saline environments may lead to the discovery of potential PGPRs that are tolerant of the abiotic stresses. A successful identification of such PGPR can potentially increase yield and ultimately efficacy of the plant extracts for *L. frutescens* that are exposed to abiotic stress. Therefore, the challenge to fulfil the ever-increasing demand of *L. frutescens* raw material would be partially realized. The identified PGPR would be employed to enhance the phytochemical profile and yield in cultivated *L. frutescens*, including those subjected to abiotic stress. Ultimately, the superior symbionts of *L. frutescens* may be used to develop biological inoculants for the plant and possibly, many other medicinal plants.

Research that focuses on individual phytochemical profile of *L. frutescens* under different environmental conditions will provide valuable information, settling many speculations used to explain variations observed in the plant. There has never been a study to establish the response of triterpenoid glycosides, flavonol glycosides and pinitol to abiotic stress. This is despite the unquestionable information that attribute the medicinal properties of *L. frutescens* to these phytochemicals.

The detection of triterpenoid glycosides and flavonol glycosides has been proposed as a tool for predicting the origin of *L. frutescens* raw materials and/or subspecies. Thus, the presence or absence of Sutherlandiosides in *L. frutescens* could be used to distinguish between subspecies or origins of plant raw materials. More focused research will be necessary to validate the proposal. It would also be interesting to compare the biological activities of the subspecies to ascertain which phytochemicals are responsible for which activity. Furthermore, if indeed the WC species are characterized by pronounced Sutherlandins and NC species characterized by Sutherlandiosides, this attribute may be taken advantage of to prescribe province-based subspecies for certain treatments. For instance, the NC subspecies may be used for treatment of cancer patients, because Sutherlandiosides exhibit superior anticancer activities.

## Figures and Tables

**Figure 1 plants-12-00196-f001:**
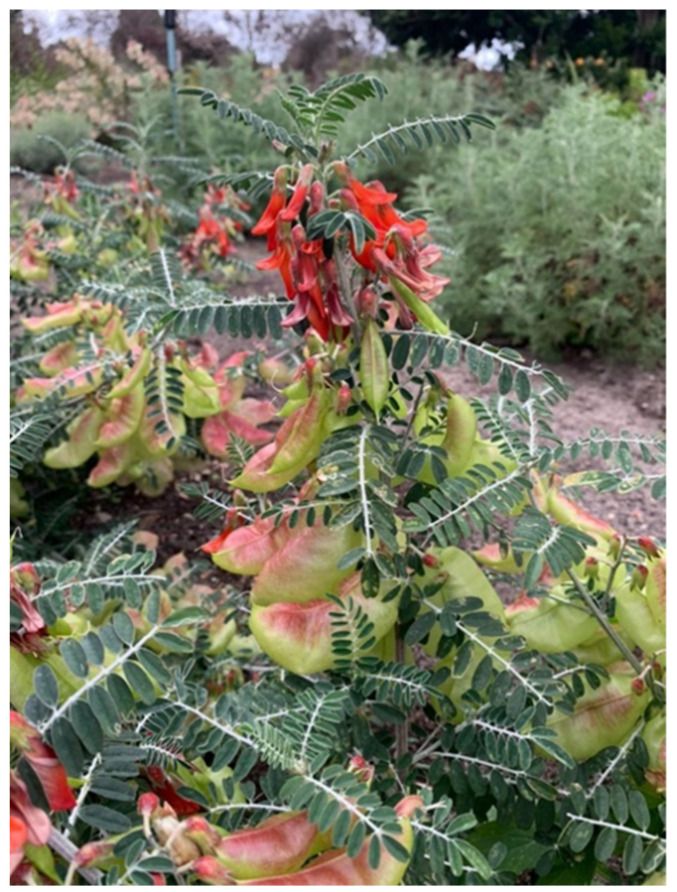
*Lessertia frutescens* at the Harold Porter Botanical Gardens, Betty’s Bay, South Africa. The bright orange–red flowers and green seed pods with shade of red are depicted. Photographer: MM Hlongwane.

**Figure 2 plants-12-00196-f002:**
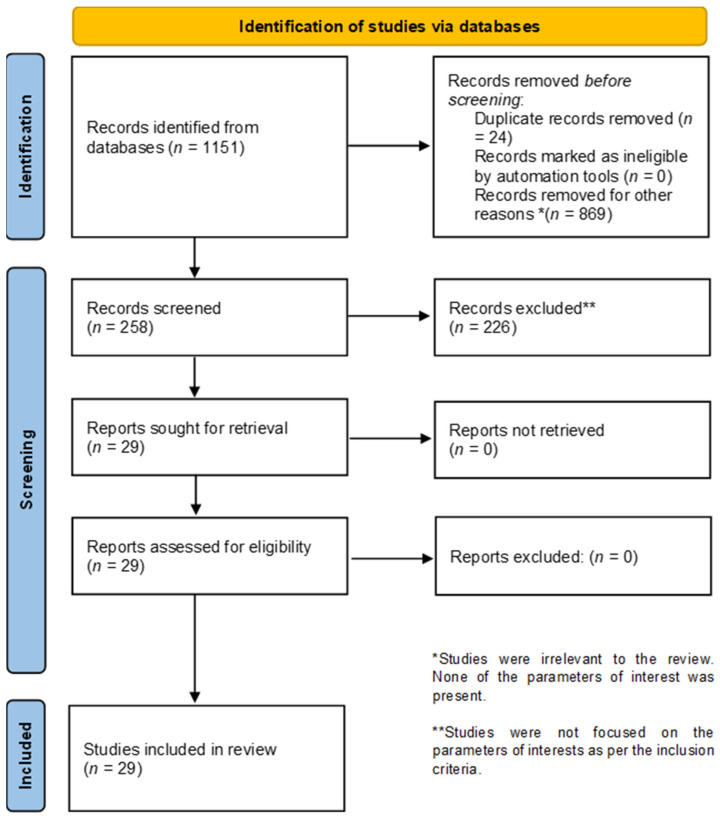
Preferred Reporting Items in Systematic Reviews and Meta-Analyses (PRISMA) flow diagram for the process followed during the review, highlighting the number of studies identified, excluded and included.

**Figure 3 plants-12-00196-f003:**
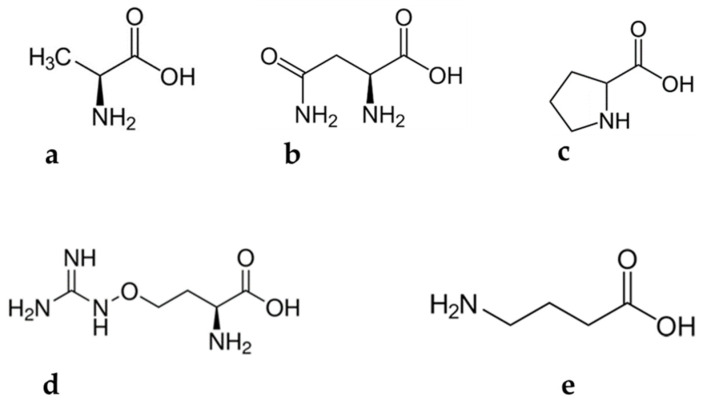
Amino acids found in *Lessertia frutescens*: (**a**) alanine, (**b**) asparagine, (**c**) proline, (**d**) canavanine and (**e**) gamma aminobutyric acid.

**Figure 4 plants-12-00196-f004:**
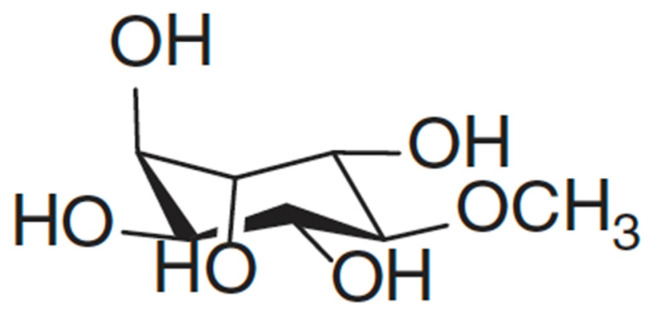
Structure of D-Pinitol adapted from Mncwangi and Viljoen [12]. CC BY license (https://creativecommons.org/licenses/by/4.0/).

**Figure 5 plants-12-00196-f005:**
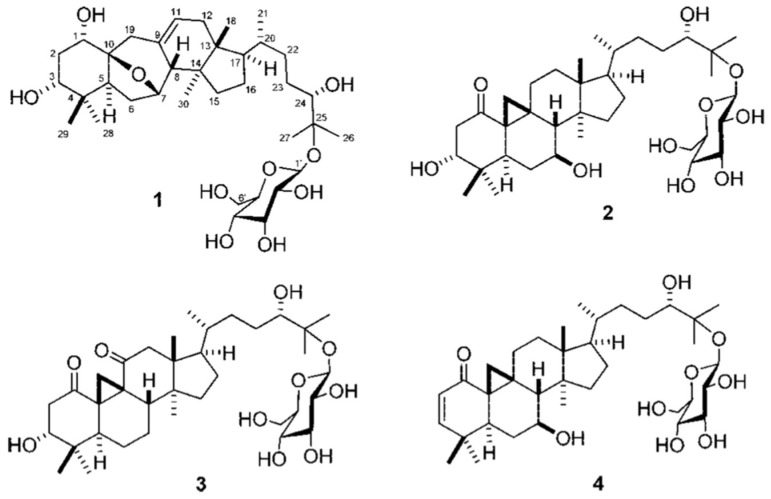
Sutherlandiosides A–D (**1**–**4**) Reprinted with permission from Ref. [73]. 2022, ACS Publications.

**Figure 6 plants-12-00196-f006:**
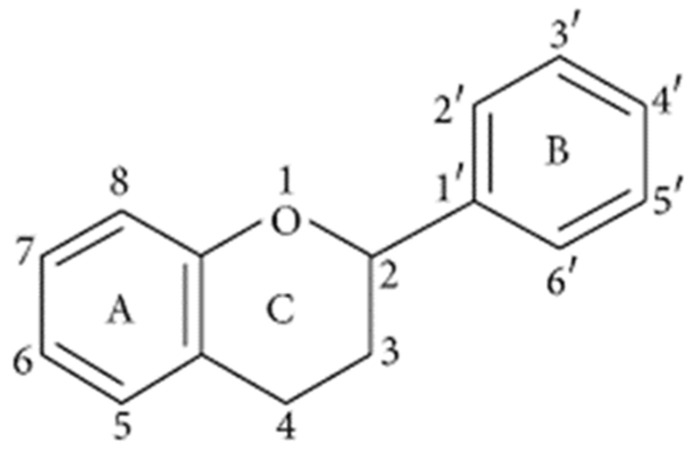
Basic structure of flavonoids. Reprinted from Ref. [76]. CC BY-NC-ND license (http://creativecommons.org/licenses/by-nc-nd/4.0/).

**Figure 7 plants-12-00196-f007:**
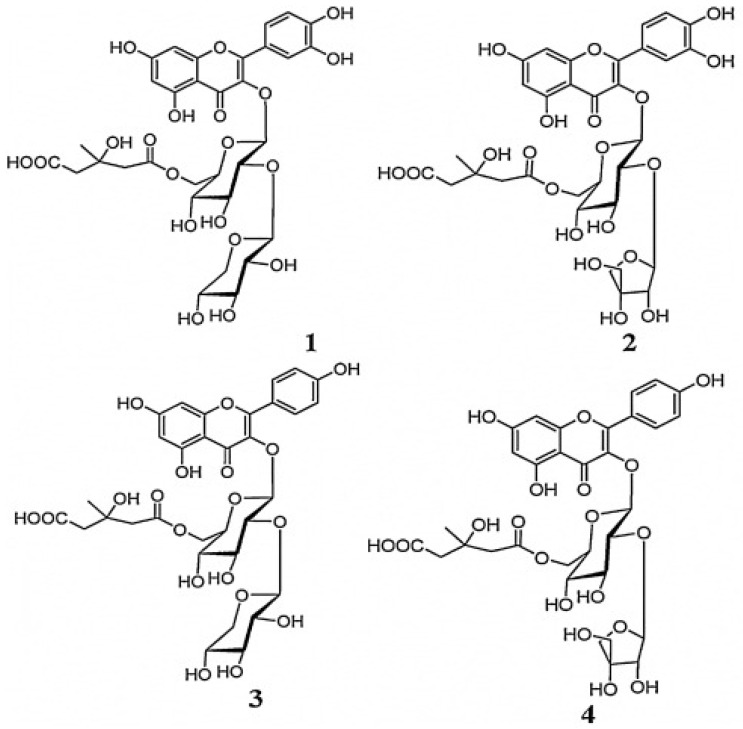
Sutherlandins A–D (**1**–**4**). Reprinted with permission from Ref. [81]. 2022, Elsevier.

**Table 2 plants-12-00196-t002:** Amino acid concentration in *L. frutescens*. Data from Moshe [33] and Mncwangi and Viljoen [12].

Amino Acids	Moshe [33], Natural Population (mg/g)	Mncwangi and Viljoen [12], Natural Population (mg/g)
Alanine	0.00–10.7	0.37–1.67
Asparagine	3.16–60.8	3.59–10.4
Canavanine	2.04–17.9	1.29–13.6
GABA	0.00–4.60	0.00–0.98
Proline	0.21–36.8	0.32–8.09

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
