# Peer review of "The Potential of Rhizobacteria to Mitigate Abiotic Stress in Lessertia frutescens"

_plants, 2023, doi:10.3390/plants12010196_

Round 1
Reviewer 1 Report
The authors proposed the following title of manuscript “Plant growth promoting rhizobacteria: A potential tool for alleviation of salt and drought stress in Lessertia frutescens”, however only section no. 8 - “PGPR for alleviation of drought and salt stress in medicinal plants”, the most corresponds to the proposed subject of this review publication.
In my opinion, this chapter should be completed and the subsequent chapters should be placed further in the manuscript.
My other comments:
Line 94. The Authors should inform on range of time of publications. Moreover, information about kind of publication (review and experimental papers etc.) must be added.
Line 108. Please, explain the inclusion and exclusion criteria, which have been used to selection of the publications to review.
Line 115. All Section 5. “Phytochemical composition of Lessertia frutescens”:
a) It should be noted which part of the plant was the source of the raw material for phytochemical analyzes.
b) The type of extracts in which the compounds were determined should be indicated.
The citation system of the bibliography is incorrect.
Line 529 – The authors should check the complete of the publication list
Lines 515 - 553 - Forni et al., Glick et al., Gopalakrishnan et al., Egamberdieva et al., Abeer et al. - are lacking in the list of bibliography
Reviewer 2 Report
Interesting topic. However, some necessary corrections are needed.
For presentation of the amino acid structures (well known, from the other hand) Authors should provide one combined Figure (Figures 4-8, and eventually also Fig. 9, should be presented as one with subpoints such as; a,b,c,…). This combined figure caption could be; “Amino acids and pinitol found in L. frutescens; A – alanine, B-asparagine, C- …”
References should be cited in square brackets as numbers, not author names. Cited references should be cited starting from the firstly released to the newest. (see; lines; 45/47; 56/57, 62/63, 121/123, 136/137, 158/159, 215/216 …etc.) Researchers findings should be cited from the earliest to the newest. Text and and also Table 1; should be rearranged this way.
Lines; 47-49 – not clear, it should be maybe; “According to Shaik et al. (2008) L. frutescens would be a beautiful ornamental plant, but its medicinal use has made it mainly used for this purpose”.
Lines 51-52; Caption for Figure 2; it should be; “Lessertia frutescens at the Harold Porter Botanical Gardens, Betty’s Bay, South Africa. The bright orange – red flowers and green seed pods with shade of red are seen.”
Line 53; constituents should be listed in order according to e.g. the type of chemical compounds; so it should be e.g. ; “…include asparagine, arginine, canavanine, gamma butyric acid (GABA), pinitol, flavonoids, including flavonoid glycosides (sutherlandins A-D), terpenoids, including terpenoid glycosides (sutherlandiosides A-D)…” The same problem in the Lines 119-121
Line 58; remove “However”
Line 60-61 it should be; “From the many phytochemicals in the plant, canavanin has gained the most attention from researchers due its ability to inhibit tumor proliferation…”
Lines 72-74; not clear; maybe it should be; “The observed by authors variability in the biological activities could be attributed to the possible effects of prevailing biotic and abiotic factors on plants phytochemical profiles….”
Line 188; “…found in the (other ?) (whole ?) aerial parts…” – please specify
Lines; 198-201; should be “Another in vivo study on rat colon carcinoma culture cells was conducted few years after…”
Lines 212-214; It should be; “… interventions on the cultivation level to optimize composition of the phytochemical compounds …”
Line 219; original research from 1949 year should be cited.
Lines 234-236; This fragment is confusing – please rewrite it.
Lines 304-306 – The scientific article shouldn’t lay on “speculations”. And this sentence is not clear; “Although speculative, the origin of the field leaves used by Shaik et al. (2011) could possibly be the same as where Moshe (1998) collected their sample for pinitol analysis.”
Lines 315-322 – be more precise. For some inspiration see as follows;
(“Triterpenes are naturally occurring alkenes included in the broad and structurally diverse group of triterpenoids, which also include natural degradation products, oxidation, and hydrogenation products. Oxygen derivatives of triterpenes are known as triterpenoids. Triterpenes and triterpenoids are slightly different from each other. Triterpenes are a diverse group of natural compounds having molecular formula C30H48)”,
and “The triterpenoids can be divided into two main classes: the tetracyclic compounds and the pentacyclic compounds. Also, in the later stages of biosynthesis, small carbon fragments may be removed to produce molecules with less than 30 carbon atoms, for example, the C27 steroids.”
Line 322; The reference cited (Parmar et al, 2013) is not recent , as we have already 2022 year; and “In fact, over 4000 natural triterpenoids have been isolated, and more than 40 skeletal types have been identified. - see “Studies in Natural Products Chemistry” – Chapter 12; Volume 67, 2020, Pages 411-461.
Line 324 and 327; should be “triterpenoid glycoside or triterpenoid glycosides” not “triterpenoids glycosides”
Line 329; should be “anti-proliferative properties”
Line 344-346; should be; “Antiviral activity tests exhibited only for sutherlandioside B a slight activity against Epstein–Barr virus (EBV), although at a potency inferior to that of the positive (?) control used in the investigation”.
Line 383; not clear ; “C3 ring” (?)
Lines 556-558; should be ; “Any reduction in amino acids profile would be of detriment to the medicinal capabilities of the plant because, on the basis of previous research knowledge, they participate in L. frutescens health benefits.” – or something like that.
Reviewer 3 Report
Scientific names as Sutherlandia frutescens, Vigna unguiculata and other must be in italics including in references.
PHYSIOLOGICAL ROLE OF ASPARAGINE AND RELATED SUBSTANCES IN NITROGEN METABOLISM 681 OF PLANTS must be in title form
Reviewer 4 Report
After reading the manuscript entitled “Plant growth promoting rhizobacteria: A potential tool for alleviation of salt and drought stress in Lessertia frutescens” seems interesting and can be published after some minor modifications.
Focus should be given to the interaction between rhizobacteria and abiotic stress in the introduction part, whether it is beneficial to increase the secondary metabolites or biological activities.
Please check line no 35.
Section 6: Although the authors mentioned the limitations of the reference materials related to the topics. However, I suggest adding a table on the effect of different abiotic stress on secondary metabolites of L. frutescens.
Reviewer 5 Report
Journal: Plants (ISSN 2223-7747)
Manuscript ID: plants-1982564
Type: Review
Title: Plant growth promoting rhizobacteria: A potential tool for alleviation of salt and drought stress in Lessertia frutescens
Lessertia frutescens is a much-respected and long-used medicinal plant that is also an attractive garden plant, and has been cultivated in gardens for many years for its fine form, striking colour and luminous flowers. It was previously known as Sutherlandia frutescens.
The present manuscript topic is investigated in the literature, and there is a very few of reference published. However, this paper gives significant contribution to the current knowledge in related field. The data are sound and it deserves to be published, after major revisions.
Title:
As present title is not making a suitable scientific sense, first delete the sign (:) and also kindly revise the title.
Abstract:
Page# 1, Line # 22; Kindly replace “Therefore, this review seeks to explore” with “Therefore, the present review will highlights”
Keywords:
Keywords should not be the same as mentioned in the title or abstract.
Introduction
Page# 2, Line # 41; Figure 1. Geographical distribution of L. frutescens in South Africa (http://pza.sanbi.org/sites/de-fault/files/info_library/sutherlfrut.pdf).
Copyright permission must be submitted to editor and properly cited with published or source reference
Page#2, Line #51; Figure 2. Lessertia frutescens with bright orange-red flowers and green seed pods with a shade of red, at the Harold Porter Botanical Gardens, Betty’s Bay, South Africa.
Copyright permission must be submitted to editor and properly cited with published or source reference
Authors completely failed to develop the hypothesis with reference to title and objective, in the introduction section.
The text has many typing and grammatical errors, capitalization issues.
English style and language requires a profound revision. However, the readability of the manuscript needs to be improved, preferably carefully reviewing by a native English speaker.
All proper nouns must be abbreviated.
Abbreviations must be described completely at first mention with brackets.
Don’t start a sentence with an abbreviation here.
Methodology:
Very much brief and short. It’s not upto the mark for a review article.
The authors of present review are asked to find and read the related published reviews in Plants MDPI Journal for proper formatting.
Results:
Collected information is sound one. It deserves to be published after major improvements.
Use www.turnitin.com to find and eliminate unnecessary self-repetition and any copied text.
Very Minute Scientific Debates.
Please cite Figure No. or Table No. in brackets at suitable places for a better connectivity in results and discussion sections as to facilitate the reader.
It has been observed that the manuscript is submitted in speedy way, without reading the instructions completely.
I would have expected slightly greater discussion of how exactly Lessertia frutescens growth was affected; more detail on the mechanisms and logical reasoning is required. There is much more scope here for discussing the implications of what these results mean.
Conclusions and prospects:
Novelty of this research work is again questionable with reference to practical significance and economic feasibility must be worked and mentioned.
References:
A few very old references have been used. These must be updated with recent research findings or removed.
Proper formatting is questionable. It must be according to MDPI Plants Journal. References formatting are inconsistent. A few DOI missing.
Verify each reference from original source and cross check references in the text and reference section.
Round 2
Reviewer 1 Report
The manuscript title has been completed and now it correlates with the content of the manuscript. The authors revised the manuscript accordingly and responded to the reviewer's comments.
Author Response
Dear Reviewer 1,
Thank you for your encouraging feedback. Your detailed review enabled us to improve the manuscript to your satisfaction and better version. We are truly grateful.
Reviewer 5 Report
Point 5:Methodology:
Very much brief and short. It’s not upto the mark for a review article.
(In revised manuscript, this section is still found poor, kindly improve it scientifically and logically, either you may quote published citations).
